# Parathyroid Cancer: A Review

**DOI:** 10.3390/cancers11111676

**Published:** 2019-10-28

**Authors:** Nikita N Machado, Scott M Wilhelm

**Affiliations:** 1Department of Surgery, University Hospitals Conneaut, Conneaut, OH 44030, USA; nikita.n.machado@gmail.com; 2Department of Surgery, Endocrine Surgery, University Hospitals Cleveland, University Hospitals, Cleveland, OH 44106, USA

**Keywords:** parathyroid cancer, epidemiology, diagnosis, surgical management, surveillance

## Abstract

Parathyroid cancer is one of the rarest causes of primary hyperparathyroidism and tends to present with more severe symptoms than its more benign counterparts. This article details various aspects of the disease process, including epidemiology, clinical presentation, and a step-wise diagnostic process for parathyroid cancer. This includes laboratory assessments as well as a proposed staging system. The en bloc principle of surgical intervention is detailed, as well as the current role of adjuvant treatments. A general guide to surveillance and the natural history of the disease is also outlined.

## 1. Introduction

Parathyroid cancer is one of the rarest causes of primary hyperparathyroidism (~1%) (the most common causes being single parathyroid adenomas, followed by double adenomas and parathyroid hyperplasia), and tends to present with more severe symptoms of hypercalcemia than its benign counterparts [1,2]. It was first described by Sainton and Millot in 1933 [3]. Parathyroid cancer is the least commonly seen endocrine malignancy worldwide. Similar to other causes of primary hyperparathyroidism (PHPT), parathyroid cancer induces symptoms by an excessive, autonomous release of parathyroid hormone (PTH) into the bloodstream.

This chapter aims to review the epidemiology of parathyroid cancer including demographics, the clinical presentation and diagnostic algorithm, surgical management, the role of adjuvant treatment and surveillance options.

## 2. Epidemiology

A study using the population database SEER (Surveillance, Epidemiology and End Results) compiled data of parathyroid cancer patients in the USA between the years of 1988 to 2003, in order to better identify changes in incidence rate, tumor parameters and treatment options over time [4]. The incidence of parathyroid cancer cases was evaluated over a period of sixteen years, and it was shown to have increased by 60% (3.58 per 10,000,000 population in the first 3 years of the study to 5.73 per 10,000,000 population in the last 3 years). In addition, the study database also found a decrease in the number of patients with larger tumors (>4 cm) and an increase in the percentage of patients with negative lymph nodes. These findings indicate an earlier point of diagnosis for these patients.

There is no sex predilection for parathyroid cancer, as opposed to primary hyperparathyroidism where there is a female preponderance (F:M ratio of 3–4:1). There has also not been any racial predilection noted for the occurrence of parathyroid cancer.

Patients with parathyroid cancer tend to be relatively younger (about 50 years at diagnosis) than the usual patient with benign PHPT [5].

## 3. Etiology and Pathogenesis

The etiology of parathyroid cancer is an interplay of both genetic and environmental factors. Studies have found an increased risk of benign parathyroid disease in patients who have been exposed to childhood radiation (especially head and neck), as well as concurrent thyroid disease, however it is not clear to what degree this is a causative factor for parathyroid carcinoma [6,7].

Recent basic molecular studies have greatly helped our understanding of the genetics behind parathyroid carcinoma, owing in part to the clinical and molecular characterization of HPT-JT (Hyperparathyroidism-Jaw Tumor), a rare autosomal dominant familial cancer syndrome. These patients develop primary hyperparathyroidism due to a combination of benign and malignant parathyroid tumors. They also are prone to developing bony tumors of the maxilla/mandible, renal cysts, and uterine tumors [8,9,10].

More recently, germline mutations have been noted in the CDC73 gene, previously known as the HRPT2 gene (which codes for the protein parafibromin) in >50% of HPT-JT kindreds, as well as in about 20% of patients thought to have sporadic parathyroid cancer. 

Parafibromin is a nuclear protein whose main function is to regulate cell transcription, and the overexpression of this particular protein causes inhibition of cell proliferation and cell cycle arrest [11].

Furthermore, gene studies on HRPT2 have shown that both somatic and germline mutations are associated with most of these tumors, and HRPT2 functions as a tumor suppressor gene [12,13].

Unlike CDC73, other germline disorders that predispose to parathyroid adenomas, such as those involving mutations in MEN1, FHH, CASR, CGM2, are rarely, if ever, associated with parathyroid carcinoma [14]. Several other mutations have been implicated in the pathogenesis of sporadic parathyroid cancer [15,16]. These genes include retinoblastoma (Rb), p53, (breast carcinoma susceptibility (BRCA2) and cyclin D1/parathyroid adenomatosis gene 1 (PRAD 1) genes [11,16] as well as epigenetic modifications, such as DNA methylation, histone modifications, microRNA misregulation [17] and circular RNA [18]. A recent study analyzed 31 parathyroid carcinomas and identified novel candidate drivers in genes that mediate chromosome organization, DNA repair, and cell cycle, as well as others that regulate MAPK signaling and immune response [19]. Another study identified mutations in genes that may be targets for therapy, such as *PTEN*, *NF1*, *KDR*, *PIK3CA*, and *TSC2* [20].

## 4. Clinical Presentation

While there is definitely an overlap between the symptomatology of benign and malignant parathyroid disease, certain findings increase the likelihood of parathyroid cancer. The following characteristics are more likely to reflect parathyroid malignancy:A higher frequency of symptomatic hypercalcemia. These patients can present with a myriad of symptoms, including nausea, vomiting, abdominal pain, constipation, fatigue, myopathy, disorientation and neurocognitive deficits;Very high serum PTH concentrations (5–10 × the upper limit of normal, as well as absolute PTH levels >500 mg/dL;Serum calcium levels >14 mg/dL;Presence of a parathyroid crisis;Presence of a palpable neck mass.

In addition to this, patients with parathyroid cancer tend to have larger tumors, as well as an increased presence of combined bone and kidney disease. In most patients with parathyroid cancer, the disease follows an indolent course marked by significant morbidity and mortality related to hypercalcemic symptoms, as opposed to tumor spread. Most treatment modalities are therefore aimed at ameliorating the signs and symptoms of hypercalcemia in advanced cases. 

### 4.1. Atypical Presentations 

While the above findings have been noted in most cases of parathyroid carcinoma, occasionally patients may remain normocalcemic. They often do present with a neck mass in these cases. There are also ~30 case reports of non-functioning parathyroid carcinomas, although this presentation is extremely rare [21,22]. Due to the difficulty with diagnosis in these cases, they tend to present at a more advanced stage of disease, and may also have more aggressive tumors [23]. These tumors have a tendency to metastasize to multiple locations including cervical lymph nodes, lungs, liver and bone. The overwhelming tumor burden is the more common reason for death in these patients, as opposed to hypercalcemia [24].

### 4.2. Hypercalcemic Crisis

This is defined as a rapid onset serum calcium level >14 mg/dl (once corrected for serum albumin levels), along with systemic signs and symptoms that correlate with the severity of hypercalcemia. 

Oftentimes, these patients can present with evidence of mental status changes (altered sensorium, lethargy, and stupor). These symptoms indicate the need for a more aggressive plan of treatment.

The first step in management is related to both volume expansion and medications to help decrease serum calcium concentrations. The following medications are commonly used: Normal saline is the intravenous fluid of choice for resuscitation and volume expansion. The initial rate of administration is between 200 and 300 cc/hr, which is titrated to ensure a urine output of 100–150 cc/hr [25].While loop diuretics aid with calcium excretion, they are not usually recommended in the absence of cardiac or renal failure, since there are possible complications and due to the presence of alternative medications that inhibit bone resorption (usually are the main proponent of hypercalcemia);Another effective medication is salmon calcitonin (4 international units/kg) with repeat serum measurements a few hours later. The goal is to evaluate for an appropriate decrease in calcium level, at which time repeat doses can be given for 6–12 h (4–8 IU/hour). It is important to note that sometimes patients develop tachyphylaxis to this medication after repeated doses, therefore the medication is not usually continued beyond 24–48 h [26];Another class of medications that are useful include bisphosphonates. These medications are non-hydrolysable compounds that adsorb to bone surfaces and inhibit calcium release by interfering with bone resorption [27];Zolendronate (4 mg IV over 15 min) or pamidronate (60–90 mg/2 h) can also be administered, with repeat doses as needed every 3–4 weeks. Zolendronate has been found to be more effective in cases of malignancy induced hypercalcemia than pamidronate;Certain patients are unable to receive bisphosphonates due to renal impairment. In these cases, a medication called denosumab may be administered instead, in addition to calcitonin. The initial dose is 60 mg subcutaneously, repeated for clinical response. [28]. Recent studies even support a higher dose regimen of 120 mg of denosumab, given every 4 weeks, which has shown to be very effective in controlling hypercalcemia from bone metastases in advanced cancer [29]. Another study included doses on day 8 and 15 during the first month, to expedite the drop in calcium and achieve a steady state of denosumab at a faster rate [30].

In addition to all these measures, it is important to avoid any supplements or foods containing calcium as well as vitamin D.

These patients require both intravenous fluid resuscitation as well as medical management prior to an expedited parathyroidectomy. 

There is a small subset of patients that have even higher serum calcium levels (as high as 18–20 mg/dL) who present with very severe, symptomatic hypercalcemia. These patients may even need hemodialysis to rapidly correct their serum calcium levels, as long they are hemodynamically stable [31].

The index of suspicion for parathyroid cancer should be higher in these patients.

What follows below is a table to compare and contrast the variable presentations in parathyroid cancer, as enumerated in Table 1. This also takes into account the variable serum calcium levels with which these patients can present.

## 5. Diagnosis

As enumerated under clinical presentation, certain findings are more likely in parathyroid carcinoma than in benign disease, including severity of hypercalcemic symptoms. However, there is no definitive laboratory diagnosis to distinguish between the two conditions.

The only definitive way to diagnose parathyroid cancer clinically is by the presence of metastatic disease, however this is usually not seen at the time of presentation [32].

Patients occasionally present with a palpable neck mass, however fine needle aspiration biopsies are discouraged since histology does not differentiate between benign and malignant disease, and there is always a risk of seeding the biopsy tract [33]. The other concern with pre-operative parathyroid biopsies is that they may result in subsequent hematoma or abscess formation, as well as procedural inflammation that increases the difficulty of any subsequent operative intervention [34]. Since the artifacts created by the biopsy can mimic parathyroid cancer in an otherwise benign lesion [35], the American Association of Endocrine Surgeons (AAES) guidelines for the management of primary hyperparathyroidism concluded that biopsies should only play a role in cases where localization is difficult e.g., intrathyroidal parathyroids, or in re-operative cases [36].

Histological criteria to aid in the diagnosis of parathyroid cancer were initially enumerated in 1973 by Schantz and Castleman. Some of the relevant diagnostic features are listed below [37]:Sheets or lobules of tumor cells with interspersed fibrous bands;Mitotic figures;Necrosis;Capsular invasion;Vascular invasion.

Of these features, gross invasion beyond the capsule and vascular invasion are the best correlates with a diagnosis of malignancy [38]. Parathyroid cancer is usually confirmed histologically by the presence of vascular invasion, as shown below, in addition to various immuno-histochemical stains that verify parathyroid differentiation (Figure 1).

### 5.1. Images Courtesy of Sylvia L. Asa, Dept. of Pathology, Case Western Reserve University, OH

An immuno-histochemical panel that includes *parafibromin, galactin-3, PGP9.5*, and *Ki67* has been used in a small series to aid with the diagnosis of parathyroid carcinoma. The study showed a sensitivity of 80% and a specificity of 100% [39]. Furthermore, the addition of other biomarkers including *Bcl-2a, parafibromin, Rb*, and *p27* may play an important role in identifying atypical parathyroid adenomas which do not present with all the classic histologic features (unequivocal angioinvasion, perineural, and gross local invasion) that we associate with parathyroid carcinoma [40]. There also exist other target pathway proteins including COX-1/2 (cyclooxygenase), Gst-pi, and members of the sonic hedgehog pathway [40]. Research is ongoing as to how to therapeutically target these markers in parathyroid cancer.

One of the main issues with using the histological criteria described previously is that a large number of these findings, including fibrosis, capsular invasion, mitosis, and necrosis can also be found as a result of post biopsy artifactual interference [35]. This likely stems from the fact that a lot of these specimens are manipulated (biopsy or ethanol) with resulting fibrosis and pseudoinvasion, which bears consideration. Our hope is that future studies will combine a thorough clinical history in addition to the biochemical markers detailed above. The ultimate goal would be to eradicate the category of atypical adenomas and definitively identify them as either degenerate benign lesions or clear-cut carcinomas.

### 5.2. Atypical Adenomas

Somewhere along the spectrum of parathyroid disease (benign adenomas to parathyroid carcinomas) lie atypical parathyroid adenomas. These patients present with large parathyroid tumors with surrounding fibrous tissue, and some of these tumors have a few histologic features in common with parathyroid cancer, however not enough to make a diagnosis of malignancy [41]. A retrospective study at a tertiary referral center in New England evaluated 3643 patients with PHPT and identified 52 patients with aggressive parathyroid tumors (18 with malignancy and 34 with atypical adenomas) [38]. 

The study showed certain parameters that were different between the two groups. Parathyroid carcinoma patients showed a significantly increased tumor size (3.5 cm vs. 2.4 cm). They were also noted to have an increased mean serum calcium level and intact PTH level. These patients had a higher occurrence of hypercalcemic crisis compared with patients with atypical adenomas. Parathyroid carcinomas usually present with an indistinct capsule compared to benign adenomas and were more likely to be adherent to adjacent structures. Of note, both groups were evaluated for a loss of *parafibromin* expression, and no significant difference was noted. 

## 6. Staging

Owing to the limited available data on tumor characteristics and prognosis, a formal staging system does not exist at this time. However, the newest release of the Tumor, Node and Metastases (TNM) cancer manual from the combined American Joint Committee on Cancer (AJCC) as well as the Union for International Cancer Control (UICC) have proposed and defined specific variables to be recorded prospectively in order to develop a formal staging system in the future [42].

Some of the proposed variables that would make up the staging system have been included belowin Table 2, modified from the original text [43]. 

## 7. Surgical Treatment of Parathyroid Cancer

### 7.1. Preoperative Imaging

Neck ultrasound and sestamibi scans are the localization studies of choice in benign parathyroid disease and may be used for similar localization purposes in malignancy as well. Surgeon-performed office ultrasounds are often the first line of diagnosis, owing to their non-invasive nature and ease of use. This is, of course, contingent on the familiarity of the provider with this technique. Usually, parathyroid carcinoma is distinguished from benign parathyroid adenomas due to a larger mass size, as well as the fact that parathyroid cancers tend to be more non-homogenous, with a decrease in echogenicity. These masses also tend to present with evidence of degeneration (both cystic cavities and calcifications may be seen, as well as irregular borders). Another measurement that proves to be helpful in these cases is a measured ratio of tumor depth: width (in the case of carcinoma, this ratio is equal to/greater than 1) [44].

This is in contrast to parathyroid adenomas, which are usually solid and homogenous, with a smaller average size as well as evidence of increased vascularity. The borders tend to be regular, and the gland appears oval or bean shaped [45]. The ultrasound imaging study is also useful for evaluating the lateral neck compartments to rule out abnormal appearing lymph nodes. If abnormal, the presence of carcinoma in those lymph nodes can be confirmed with an ultrasound guided tissue biopsy.

Sestamibi scans (dual-tracer parathyroid imaging) are the other imaging modality that are commonly used in pre-operative localization. While they do have a relatively high level of sensitivity for identifying parathyroid lesions, they are not as adept at differentiating between parathyroid adenomas and parathyroid carcinoma [46]. Their mechanism of action is related to an increased uptake of technetium 99-m within the mitochondria of the abnormal parathyroid gland [47]. Another important point to note is that parathyroid cancers that have undergone cystic degeneration are likely to show up as false negative sestamibi scans [48]. In addition, there is occasionally uptake within the thyroid gland itself, which can lead to false positive results [49]. The variation in these results is why sestamibi scans are often accompanied by another imaging modality, whether that be ultrasound or CT/PET imaging.

In cases where parathyroid cancer is suspected based on preoperative labs and symptoms, higher resolution imaging, e.g., 4D CT/MRI may also be helpful to identify possible ectopic glands and relationship to surrounding soft tissue structures [50]. MRI also has a role in those patients where surgical clips have been placed during the initial operation and imaging is being performed for recurrent disease, as it avoids scatter and distortion unlike CT scans [51].

18-FDG PET (fluoro-deoxy-glucose positron emission tomography) has now emerged as a useful modality in the detection of tumors, especially malignancies. This effect is mainly due to the fact that malignant cells have an increased glucose metabolism due to the presence of more glucose transport proteins [52]. While primary tumors are fairly easily evaluated with PET imaging, it is important to note that micrometastatic lesions (<6 mm) can be missed by a PET scan when it is performed as an individual exam. PET scans are primarily qualitative in nature, and they can be somewhat quantified using standardized uptake values (SUVs). Certain studies have evaluated the association of an elevated SUV level with the aggressiveness of the parathyroid tumor and found a positive correlation [53]. While PET imaging has a role in the identification of loco-regional spread in the case of certain primary parathyroid cancers, it plays an even more important role in parathyroid cancer recurrence for restaging purposes.

One of the limitations of PET scans is the fact that false positive results can arise even in the setting of infection or inflammation [53]. This is more important in post-surgical patients, who may have evidence of either acute or chronic inflammation after surgery. A post-treatment infection or inflammatory lymphadenopathy can also show up as a false positive on a PET scan. For this reason, PET scans after treatment are usually delayed for 3-6 months to reduce the likelihood of false positive results. Of course, sites that appear suspicious on PET scans during restaging should be confirmed with other imaging studies with or without the addition of tissue diagnosis to distinguish them from an inflammatory/infectious process [53]. This is the reason these studies are often performed in combination (e.g., PET-CT), to improve diagnostic efficiency.

### 7.2. Operative Intervention

The mainstay of treatment in the case of parathyroid carcinoma is surgical. Excision of the tumor is performed in two different scenarios. Surgery can be performed as the initial procedure at the time of diagnosis, as well as in the case of excision for recurrent or metastatic disease.

The best chance for cure is with a complete en bloc resection, especially when the diagnosis of parathyroid carcinoma is suspected preoperatively. In this case, preoperative imaging with CT/MRI in addition to the USG/sestamibi scan may help evaluate contiguous structures with potential tumor involvement.

The status of vocal cord mobility may also be very helpful if garnered preoperatively, as it helps to evaluate any potential extent of recurrent laryngeal nerve involvement. In some cases this significantly changes the extent of the planned operation.

#### Principles of en Bloc Surgery

A complete and thorough exploration of all four glands will help identify the presence of concurrent adenomas and carcinomas. While rare, multiglandular carcinomas have been described in the literature [54];A bloodless field and meticulous attention to detail ensures no inadvertent injury to neighboring structures;Minimal manipulation of the tumor itself to avoid rupture of the capsule and tumor spillage. This is usually accomplished by removing the ipsilateral thyroid lobe in toto with the tumor [55,56];Careful inspection for any tumor encroachment into the surrounding strap muscles or other structures, which may have to be resected with the mass. The most common structures affected by local tumor invasion are the ipsilateral thyroid lobe, ipsilateral strap muscles, ipsilateral recurrent laryngeal nerve, esophagus, and trachea [56];Nodal involvement necessitates a regional lymph node dissection of that compartment. It is important to note that prophylactic lateral neck dissection has not been shown to improve survival, is associated with an increased morbidity, and is therefore not recommended [57];In most cases, the recurrent laryngeal nerve can and should be preserved. However, if there is evidence of RLN involvement, it may be sacrificed and resected with the tumor [58].

On gross inspection, the tumor usually appears to have a firm or hard consistency as well as a dense fibrous capsule, which gives it a white or greyish-white hue [59]. On the other hand, adenomas tend to be soft and reddish-tan in color, as well as well-circumscribed. As described above, it often adheres to nearby structures and it is this feature that usually aids the surgeon in making the diagnosis of parathyroid cancer. As shown in the following pictures, Figure 2 demonstrates the intra-operative appearance of parathyroid cancer, while Figure 3 shows the operative specimen including the parathyroid cancer and ipsilateral thyroid lobe.

Intraoperative PTH monitoring is as crucial in malignancy as it is in operations for benign conditions. If the serum PTH levels drop to <50% of the pre-operative values within 10 minutes of removal of the affected gland, it is reasonably assumed that most, if not all of the disease has been eradicated [60].

If, however, the levels stay elevated, there arises concern for either residual or metastatic disease. At this point, it may be prudent to close and obtain further imaging and localization studies in the post-operative period.

The importance of correct technique in en bloc resection cannot be overstated. Studies have indicated that patients who were diagnosed preoperatively with parathyroid cancer and underwent an en bloc resection had a recurrence rate of 33%, compared to those who were diagnosed after their initial surgery. These patients had a recurrence rate of >50% [61]. This demonstrates that a proper oncologic resection often follows a high index of clinical suspicion based on pre-operative diagnostics. For this reason, it is important to consider referring these patients to experienced endocrine/head and neck surgeons with a high volume of experience, who would be well placed to make the distinction between benign and malignant parathyroid pathologies and plan their operative approach accordingly.

Another controversy in the surgical management of parathyroid carcinoma revolves around the decision to perform a prophylactic central lymph node dissection. Lymph node involvement in parathyroid cancer is relatively more infrequent compared to recurrence in other neighboring soft tissue structures within the neck [62]. Evaluation of some retrospective data has not necessarily shown a difference in the rate of metastases or death in patients who have their lymph nodes examined at the time of surgery. On the other hand, a sensitivity analysis performed in the same study did show an increased likelihood of positive lymph nodes in tumors larger than/equal to 3 cm in size [63].

### 7.3. Post-Operative Management

All patients require close monitoring (both clinical and laboratory) after surgery. This especially applies to serum calcium levels, since most of these patients are highly susceptible to hungry bone syndrome. This may be exaggerated in patients who received preoperative bisphosphonates for treatment of a hypercalcemic crisis. These patients may require a combination of IV and oral calcium, as well as calcitriol. The more severe cases may require extended hospitalization with access to a continuous IV calcium infusion until their calcium stores are adequately repleted [64]. At this point, most patients can be discharged on a normal diet with minimal oral supplementation. They require regular monitoring of their calcium and PTH levels every 3 months for up to one year to evaluate for recurrence [11]. If the levels remain stable at this point, the duration between laboratory assessments can gradually be increased.

## 8. Adjuvant Therapy for Parathyroid Cancer

### 8.1. Radiotherapy

Traditionally, parathyroid carcinoma is known to be fairly radio-resistant. However, small patient series from multiple hospitals in the last few years have shown lower recurrence and longer disease free survival with the use of adjuvant radiotherapy [65,66,67]. A retrospective review of 16 patients with parathyroid carcinoma at the Princess Margaret Hospital in Toronto evaluated the long-term response to adjuvant radiotherapy vs. none in patients who had undergone surgery. The study noted a 5- and 10-year disease specific survival rate of 100% and 69% in those who had both surgery and radiation, vs. 80% and 43% in those patients who had only undergone surgery. There was also an improvement in both locoregional and distant disease control rates [67]. Understandably, given the low incidence of this disease, large scale studies to replicate these results may not be feasible.

Similar to other neuroendocrine tumors, parathyroid cancers have also been found to express somatostatin receptors (SSTs). Immunohistochemical studies have shown a prevalence of SST 1–5 within either the cytoplasm or nucleus of parathyroid tumors [68]. The extent of SST 2–5 profile expression is variable, depending on how benign or malignant the tumor is. At this time, SST 5 offers us another potential new marker for parathyroid cancer. Similar to the fact that peptide receptor radiotherapy is being utilized in cases of pancreatic neuroendocrine tumors, there may possibly be a role for its utilization in parathyroid carcinoma, although published studies on the same are pending at this time [68].

### 8.2. Chemotherapy

Cytotoxic chemotherapy is even less commonly used than adjuvant radiation for parathyroid cancer to reduce tumor burden. Isolated case reports have shown modest success with single or combination regimens, including dacarbazine alone or in combination with 5-fluorouracil(5-FU) and cyclophosphamide [69,70].

### 8.3. Immunotherapy

Newer agents for targeted molecular and immune therapy may be the next frontier in treating parathyroid cancer. Certain experimental immunologic agents have been shown to decrease tumor size in a few isolated cases of parathyroid carcinoma [71]. This is extremely important since a decreased tumor burden also helps to decrease the associated symptoms of hypercalcemia, which is invariably the cause of the majority of recurrence related symptoms and death. Currently, biologic agents based on genes like *parafibromin*, as well as telomerase inhibitors like azidothymidine are being tested with encouraging in vitro results [70]. Isolated cases have also shown a response to metastatic disease from parathyroid cancer with sorafenib. Both referenced studies were related to lung metastases from parathyroid carcinoma, and the regression is likely related to the anti-angiogenic function of sorafenib (blocks VEGF receptors, BRAF as well as PDGFR) [72,73].

### 8.4. Agents for Symptom Palliation/Management of Hypercalcemia

Some of the adjuvant modalities of treatment are primarily being used, not so much for a curative intent, but more so for palliation. The goal, as described above, is to help with decreasing tumor burden and therefore hypercalcemia. In addition, if physically reducing the tumor burden is not an option, certain medications can be used to palliate patients by managing the associated hypercalcemia. Our current regimens include a combination of bisphosphonates and calcimimetic agents (some of which have been discussed earlier in this article). Mention must also be made of denosumab, a human monoclonal antibody whose mechanism of action is against RANKL (Receptor Activator of Nuclear Factor Kappa-B Ligand) [74]. Denosumab inhibits osteoclast function, and in that capacity was originally approved for the management of post-menopausal osteoporosis. Due to its powerful hypocalcemic effect, it is now finding a role in the management of refractory hypercalcemia, and also has additional advantages in comparison with medications like bisphosphonates. This is due to the fact that denosumab does not require any adjustment in those patients who are renally impaired, as well as the ease of administration (subcutaneous route vs. intravenous) [75]. The side effects most often seen with denosumab include bone pain, gastrointestinal symptoms like nausea and diarrhea, shortness of breath as well as a few rare cases of osteonecrosis of the jaw (which can be seen in patients who have been on this medication for a period of months) [76].

Other medications that can be considered for the management of refractory hypercalcemia include cinacalcet and gallium nitrate. The mechanism of action of cinacalcet is the decrease in PTH production, which makes it an apt choice for parathyroid cancer related hypercalcemia [77]. The initial dose is 30 mg twice daily, which can be increased after 2–4 week intervals, based on how well the patient tolerates it [42]. Gallium nitrate has been described in the literature as an option for treatment of hypercalcemia due to similar osteoclastic inhibition. It also helps to increase renal calcium clearance, and is administered as slow IV infusion of 200 mg/m2 over a period of five days [78].

In the subset of patients who do not respond to any medical management of their refractory hypercalcemia, it is important to consider hemodialysis, if their renal function does not permit aggressive hydration [26].

### 8.5. Newer Treatment Modalities

Another, relatively newer modality also includes ablation of parathyroid cancer tumor burden with ethanol. This resulted in a temporary tumor shrinkage, associated with a decrease in both serum calcium and PTH levels [79]. Further evaluation may be needed, however this could suggest a role for image guided palliation for certain cases of unresectable parathyroid carcinoma.

The treatment strategies detailed in the section above are also summarized in Figure 4 below, to provide the reader with a broad overview.

## 9. Management of Recurrent Disease

While a preoperative diagnosis of parathyroid carcinoma allows for a more complete surgical resection, a significant number of patients are diagnosed after their initial operation and therefore have not undergone a proper oncologic resection. This increases their recurrence rate to >50%, as described before. Most tumor recurrences occur 2–3 years after the initial surgery, although recurrences after >20 years have also been described [55,69].

These patients usually present with a gradual rise in their PTH and calcium levels. 

Once their hypercalcemia has been medically managed, it is imperative that good imaging studies are obtained to aid with localization of the tumor. This includes both neck ultrasounds with sestamibi scans, as well as less commonly used 4-D CT and MRI.

If these tests are inconclusive, there is a role for selective venous catheterization and sampling to assay PTH levels. This usually occurs if at least two separate complementary imaging studies have been used and the tumor is still not able to be identified [80].

Surgical management in these cases includes resection of all functional tumor tissue, including wide local excision within the neck and mediastinum. This principle also applies to metastasectomies [81], since the goal of surgery is to reduce the tumor burden and help ameliorate the symptoms of hypercalcemia [55,82]. Palliative surgeries may be done multiple times per patient, owing to the indolent nature of the disease and relatively high rate of recurrence.

## 10. Survival and Outcomes

Parathyroid cancer is an indolent, slowly progressive cancer since the tumor has a low malignant potential. Very few patients initially present with regional lymph node involvement (<5%) or with distant metastatic disease (<2%) [83]. This malignancy has a propensity for local recurrence and spreads to contiguous neck structures. Five-year survival rates as pooled from different registries and case series are in the range of 76–85% and 10 year survival between 49% and 77% [58,84].

A survey of the US National Cancer Database demonstrated a significant improvement in 5-year survival with current treatment methodology (increased from 66% to 82%) [84]. Of note, certain factors have been associated with a shorter survival, including larger tumor size, as well as male gender and an older age at the time of diagnosis. Interestingly, no significant relationship has been found between lymph node status or radical parathyroid surgery and patient survival [85].

As mentioned before, most patients do not die from tumor disease burden since it is slow growing. Instead, most patients with metastatic disease die from complications of hypercalcemia.

## 11. Summary

Parathyroid cancer remains a difficult problem to treat since it such a rare tumor, with limited effective treatment options beyond initial surgical resection. This is the reason why early identification of patients who might be at risk for parathyroid cancer in the pre-operative period is paramount. This high index of suspicion enables the surgeon to plan for an appropriate en bloc surgery, giving the patient the best chance for cure.

Future options for treatment including immunotherapy have been summarized above, and with further refining, will hopefully form an important part of our arsenal to treat parathyroid cancer.

We do know that the prevention of parathyroid cancer is not feasible, however there may be a way to mitigate the risk in some patients.

Since certain populations are at a higher risk of developing parathyroid cancer, genetic analysis may be a way to optimize clinical management and aid in surveillance for developing parathyroid carcinoma. At present, that includes germline CDC73 analysis in those with HPT-JT syndrome, malignant parathyroid histology or familial isolated hyperparathyroidism [86]. Screening for patients with HPT-JT associated cancer begins in early childhood, before patients are symptomatic. In addition to the genetic testing mentioned above, these patients also undergo biochemical testing every 6–12 months, renal ultrasound, and dental imaging every 5 years. The frequency of testing can be individualized according to the patient [85].

This also applies to family members of the individuals mentioned above, who in addition to screening for the CDC73 mutation, should also have scheduled screening for primary hyperparathyroidism [85].

Finally, future research is integral to improve our diagnosis and treatment of this disease. It would be worthwhile to pool resources of multicenter trials, as well as to consider tissue banking of these rare tumors, for further use in basic, clinical, and translational research.

## Figures and Tables

**Figure 1 cancers-11-01676-f001:**
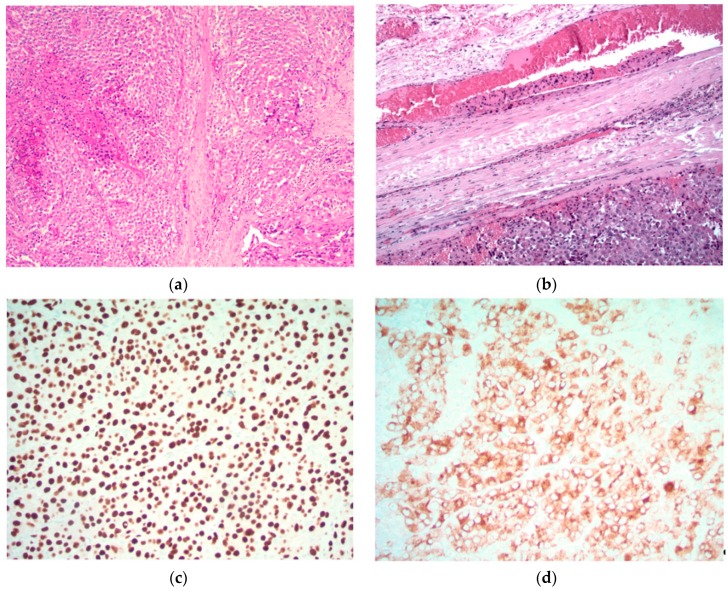
Parathyroid carcinoma is characterized by sheets of parathyroid cells with fibrosis and focal necrosis (**top left**). The diagnosis is confirmed by the identification of unequivocal vascular invasion defined as tumor cells within vascular channels associated with fibrin (**top right**). The tumor cells exhibit nuclear positivity for GATA3 (**bottom left**) and cytoplasmic positivity for chromogranin (**bottom right**) and parathyroid hormone (not shown), confirming parathyroid differentiation. Original magnification (a) ×120; (b) ×200; (c) and (d) ×400.

**Figure 2 cancers-11-01676-f002:**
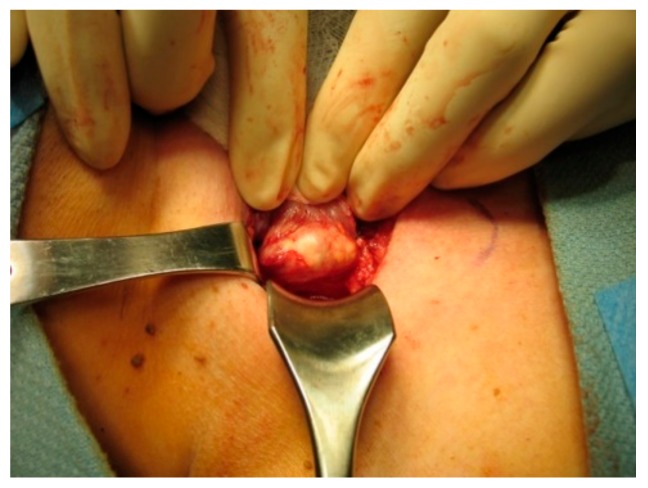
Intraoperative exposure of parathyroid cancer.

**Figure 3 cancers-11-01676-f003:**
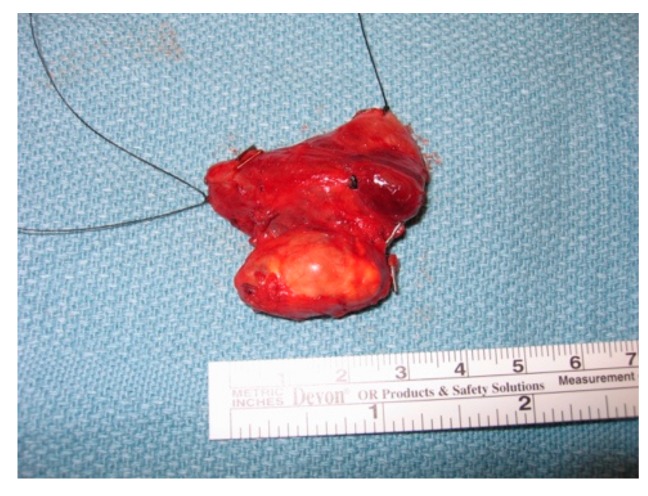
Operative specimen of parathyroid cancer with ipsilateral thyroid lobe.

**Figure 4 cancers-11-01676-f004:**
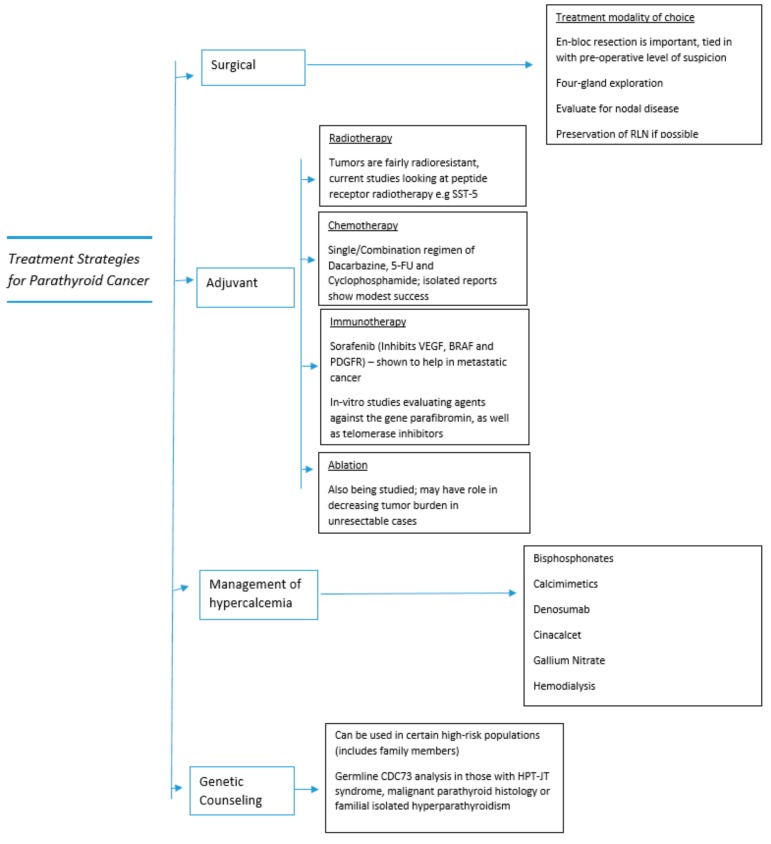
Treatment strategies in parathyroid cancer.

**Table 1 cancers-11-01676-t001:** Variable presentation in parathyroid cancer. Parathyroid hormone, PTH.

Normocalcemic (Serum Ca < 10.5 mg/dL)	Hypercalcemic	Hypercalcemic Crisis (Serum Ca > 14 mg/dL)
Can be asymptomatic, may have palpable mass	Usually symptomatic, gastrointestinal disturbances, renal stones or fatigue, neurocognitive issues	Can present in extremis, with altered sensorium or stupor
Present at more advanced stages	Should have a high index of suspicion based on calcium and PTH levels	Aggressive management with hydration, bisphosphonates, calcitonin or denosumab
Higher tendency to metastasize		Expedite surgery

**Table 2 cancers-11-01676-t002:** Variables for proposed staging system for parathyroid cancer (AJCC/UICC).

Patient Factors	Tumor Related Factors	Lab/Histological Factors
Age at diagnosis	Size of primary tumor	Highest preoperative calcium
Gender	Presence of invasion into surrounding tissue	Highest preoperative PTH
Race	Distant metastatic disease	+ lymphovascular invasion
Genetic mutations	Number of lymph nodes removed	Histological grade (high or low) and Ki67 index
	Number of positive lymph nodes	Mitotic rate
	Weight of primary tumor	Solid vs. trabecular growth pattern
	Time to recurrence	Tumor necrosis

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
