# Peer review of "Parathyroid Cancer: A Review"

_cancers, 2019, doi:10.3390/cancers11111676_

Round 1
Reviewer 1 Report
The comprehensive review by Machado et al. is covering all over parathyroid cancer, and written in detail. However, the content is a chapter of a textbook rather than a review article. One of the strongest defects is that there exist no definite conclusions and prospects for the future, which is essential for this kind of manuscript. In addition, general content is too lengthy, and focused points seems obscure.
In detail, the section of epidemiology (section 2) and demographics (3) should be combined. Hypercalcemic crisis (5.2) is not specific for parathyroid cancer, and the explanation should be shortened. The description about role of pre-operative imaging (7) or post-operative management (10) should be incorporated into the section about surgical treatment (9). The description about management of recurrent disease (11) seems disproportionate to the flow. The title of section 14 “biotherapy” is obscure, and explanations about molecular-targeted agents, ablation or agents for hypercalcemia are mixed. The significance of prevention seems undetermined, and it would be better to discuss the content as a part of future prospects including problems about cost-benefit or genetic counseling. Finally, I recommend one figure summarizing treatment strategies for parathyroid cancer to promote the readers’ comprehension.
Reviewer 2 Report
This is an excellent review regarding a rare tumour: I have only a few minor comments.
Line 67; I have certainly seen a parathyroid carcinoma in MEN1; while very rare, it is not unheard of. Line 132; denosumab is useful even with normal renal function, and may be used in patients refractory to bisphosphonates. In the same line, can the authors provide a recommendation of a dose regime for zolendronate and denosumab in such patients? For denosumab, 120mg monthly has been recommended. It would be helpful to include SI units as well.Author Response
Please see attachment.

Reviewer 3 Report
This review on parathyroid (PT) cancer is for the most part clearly written, terrifically easy to comprehend, and well organized. My comments and suggested edits will be listed in order by manuscript line number, and are mostly focused on grammar.
Line 39. The word "point" is used twice. Replace the first use with "indicate."
Line 69. There are 2 periods after citation 16.
Lines 81, 90, & 211. The authors' 2nd section describes PT cancer epidemiology, for which the term "incidence rate" is generally understood to be number of new cases over a specific unit of time. The use of the term "incidence" in lines 81, 90, 211, however, does not describe a time period. Instead the authors are presumably using the term "incidence" as equivalent to frequency, i.e., a higher frequency of symptomatic hypercalcemia. To avoid confusing readers who, after reading the Epidemiology section, may be in the frame of mind to interpret "incidence" as "incidence rate", consider substituting "frequency." Also for line 90, I might re-write this as "increased presence of [combined??] bone and kidney disease", if the authors' intent is that these 2 conditions tend to be observed as co-occurring.
Lines 142-43 and Figure 1. Line 142 refers to a table detailing differences in PT cancer presentation "...enumerated in figure 1." So is Figure 1 more of a table or a figure? I understand that the arrow over the table is a figure that indicates greater likelihood of PT cancer as rows are read left to right, but could this be written in a table header as something like "Somewhat suspicious -----> highly suspicious", thereby converting this figure into a table vs. a hybrid figure-table?
Lines 161-62: Awkwardly phrased as written. I'd omit "have also included" in line 161, and "which" in line 162. This would retain sentence meaning.
Lines 288-296. Some of these lines are complete sentences, whereas others are sentence fragments (e.g., line 293). Line 292 needs a period. Please make this section a set of bullet points as in lines 112-132, are write as complete sentences.
Line 300-1. Use a comma after "survival", omit "and" at line 301, add "and" before "therefore."
Lines 323-326. This paragraph seems to speak more about the importance of a preoperative diagnosis of PT cancer (so en-bloc resection is planned in advance) to lower recurrence rate compared with post-op diagnosis. I know the authors are aware of this issue, as noted in lines 347-348, but the issues of "proper oncologic resection" and pre-op diagnosis are highly intertwined, and the authors may want to consider stating this explicitly, and also make suggestions for preoperative caution and surgical preparation (for a possible en-bloc procedure) when PT cancer is suspected. This may also be a place to recommend patient referral to endocrine or head/neck surgeons with high PT experience who would be aware of the potential signs of PT cancer vs. other PT-related diagnoses.
Line 433. Is it acceptable to start a sentence with a number, i.e., "5 year..."? I know this may be journal specific, but I'd write "Five.."
Line 434. Omitting "shown to be" makes this sentence more clear.
Author Response
Please see the attachment.

This manuscript is a resubmission of an earlier submission. The following is a list of the peer review reports and author responses from that submission.